# Pharmacological Modulators of Autophagy as a Potential Strategy for the Treatment of COVID-19

**DOI:** 10.3390/ijms22084067

**Published:** 2021-04-15

**Authors:** Gustavo José da Silva Pereira, Anderson Henrique França Figueredo Leão, Adolfo Garcia Erustes, Ingrid Beatriz de Melo Morais, Talita Aparecida de Moraes Vrechi, Lucas dos Santos Zamarioli, Cássia Arruda Souza Pereira, Laís de Oliveira Marchioro, Letícia Paulino Sperandio, Ísis Valeska Freire Lins, Mauro Piacentini, Gian Maria Fimia, Patrícia Reckziegel, Soraya Soubhi Smaili, Claudia Bincoletto

**Affiliations:** 1Department of Pharmacology, Escola Paulista de Medicina, Universidade Federal de São Paulo (UNIFESP), 04044-020 São Paulo, Brazil; anderson.leao@unifesp.br (A.H.F.F.L.); adolfo.erustes@gmail.com (A.G.E.); indiemmorais@gmail.com (I.B.d.M.M.); talitavrechi@gmail.com (T.A.d.M.V.); lucaszamarioli@gmail.com (L.d.S.Z.); cassia.aspereira@gmail.com (C.A.S.P.); l.marchioro@unifesp.br (L.d.O.M.); leticiasperandio7@gmail.com (L.P.S.); isis.valeska.lins@gmail.com (Í.V.F.L.); reckziegel.patricia@unifesp.br (P.R.); ssmaili@unifesp.br (S.S.S.); claudia.bincoletto@unifesp.br (C.B.); 2Department of Biology, University of Rome “Tor Vergata”, 00133 Rome, Italy; mauro.piacentini@uniroma2.it; 3Department of Epidemiology and Preclinical Research, National Institute for Infectious Diseases IRCCS ‘La Zaro Spallanzani’, 00149 Rome, Italy; gianmaria.fimia@inmi.it; 4Department of Molecular Medicine, University of Rome La Sapienza, 00185 Rome, Italy

**Keywords:** COVID-19, autophagy, pharmacology

## Abstract

The family of coronaviruses (CoVs) uses the autophagy machinery of host cells to promote their growth and replication; thus, this process stands out as a potential target to combat COVID-19. Considering the different roles of autophagy during viral infection, including SARS-CoV-2 infection, in this review, we discuss several clinically used drugs that have effects at different stages of autophagy. Among them, we mention (1) lysosomotropic agents, which can prevent CoVs infection by alkalinizing the acid pH in the endolysosomal system, such as chloroquine and hydroxychloroquine, azithromycin, artemisinins, two-pore channel modulators and imatinib; (2) protease inhibitors that can inhibit the proteolytic cleavage of the spike CoVs protein, which is necessary for viral entry into host cells, such as camostat mesylate, lopinavir, umifenovir and teicoplanin and (3) modulators of PI3K/AKT/mTOR signaling pathways, such as rapamycin, heparin, glucocorticoids, angiotensin-converting enzyme inhibitors (IECAs) and cannabidiol. Thus, this review aims to highlight and discuss autophagy-related drugs for COVID-19, from in vitro to in vivo studies. We identified specific compounds that may modulate autophagy and exhibit antiviral properties. We hope that research initiatives and efforts will identify novel or “off-label” drugs that can be used to effectively treat patients infected with SARS-CoV-2, reducing the risk of mortality.

## 1. Introduction

Coronavirus disease 2019 (COVID-19) is a severe acute respiratory syndrome caused by the infectious coronavirus SARS-CoV-2 [1,2]. It was first described by the Chinese Center for Disease Control and Prevention in December 2019 as a mysterious viral respiratory disease that emerged in the city of Wuhan, Hubei province, China. On 12 January 2020, the World Health Organization (WHO) temporarily named the virus as the 2019 novel coronavirus (2019-nCoV), and by the end of the same month declared it a “public health emergency of international interest” [3]. Later, after phylogenetic analysis, this coronavirus (CoV) was officially named as the severe acute respiratory syndrome coronavirus 2 (SARS-CoV-2) [4]. On 11 February 2020, the WHO announced that the disease caused by the SARS-CoV-2 would be called “COVID-19”, and by the end of March it had already spread across five continents of the world resulting in some sort of lockdown for 1/3 of humanity. Only in 2 September 2020, WHO recommended corticosteroids as an effective treatment for seriously ill COVID-19 patients, but global deaths kept rising reaching 1 million by the end of the same month. Several other drugs were clinically used in the same effort to contain the deaths caused by COVID-19. Finally, only in December 2020, the Medicines and Healthcare products Regulatory Agency (MHRA) in the United Kingdom and the Food and Drug Administration (FDA) in the United States of America (USA) authorized the emergency use of Pfizer/BioNTech’s and Moderna’s vaccines against COVID-19 [5]. Nevertheless, worldwide vaccine plans are yet to be implemented and novel mutations of the SARS-CoV-2 are rapidly emerging [6,7] demanding continuous research on therapeutics to manage COVID-19. By the end of February 2021, the number of global deaths related to COVID-19 was close to 2.5 million.

## 2. Coronavirus: Concepts; Types; Compositions; Mechanisms of Infection and Replication

The SARS-CoV-2 is a new viral strain of the coronavirus (CoV) family. It belongs to the *Betacoronavirus* genus (β-CoVs) [8] and represents the third CoV outbreak in the last 20 years, following the severe acute respiratory syndrome coronavirus (SARS-CoV) [9,10] and the Middle-East Respiratory Syndrome coronavirus (MERS-CoV) [11]. Individuals who were previously infected with these CoVs, and those infected with SARS-CoV-2, present similar symptoms, which include dry cough, fever, headache, dyspnea and pneumonia [1,12,13]. Most patients that test positive for COVID-19 (about 80%) are asymptomatic or exhibit mild to moderate symptoms, however, approximately 15% of them progress to severe pneumonia and about 5% die due to acute respiratory distress syndrome, septic shock and/or multiple organ failure. The overall mortality rate of SARS-CoV-2 is estimated to be between 3 and 5% [1,14].

The genome of CoVs consists of a single-stranded RNA, and a notable translation product is the transmembrane spike S-glycoprotein expressed on the surface. Each spike monomer is composed of an S1 subunit and S2 subunit, which are known to bind to membrane receptors present on the surface of human cells. In the case of SARS-CoV-2, it binds to the host cell angiotensin-converting enzyme 2 (ACE2) receptor [15], the same cell surface target as SARS-CoV [2], through the action of the S1 subunit. Previous research showed that the ACE2 protein is associated to the transmembrane serine protease 2 (TMPRSS2), which cleaves S2, generating S2′ and promotes the virus–cell membrane fusion [16]. In this sense, ACE2 and TMPRSS2 are crucial for SARS-CoV-2 infection and potential pharmacological targets for COVID-19 treatment.

The ACE2 protein belongs to the membrane-bound carboxypeptidase family and is responsible for converting angiotensin II to angiotensin [1,2,3,4,5,6,7]. It is widely distributed throughout the human body, with supramaximal levels in the small intestine, testis, kidneys, heart, thyroid, adipose tissue, colon, liver, bladder, adrenal glands and lungs (mainly in type II alveolar cells) and macrophages. Lower ACE2 levels are found in the blood, spleen, bone marrow, brain, blood vessels and muscles [17]. Thus, understanding how the expression of ACE2 affects SARS-CoV-2 infection is important for the development of preventive/curative measures against infection. Analysis of the SARS-CoV-2-S protein revealed almost 80% amino-acid identity with the SARS-CoV-S protein [18].

In addition, cathepsin-mediated protein S cleavage is also a critical step for SARS-CoV-2 infection, as the acidic pH in lysosomes influences the entry of the virus into human cells [19]. Currently, it is understood that after the release of the genomic RNA into the cytoplasm, the viral replicase nonstructural proteins 3 and 4 (nsp3 and nsp4) are translated, which initiates the rearrangement of the endoplasmic reticulum (ER) membranes into double-membrane vesicles (DMVs) [20,21]. It has been speculated that the DMVs accommodate viral RNA replication [20,22,23]. Next, in the ER and Golgi intermediate compartment, the newly synthesized genomic RNA molecules are assembled into virions [24], and the infectious virions are transported to the secretory pathway where they are released by exocytosis [25]. Herein, we will discuss all the steps involved in the replication of SARS-CoVs using the autophagy machinery.

## 3. Coronavirus Hijack the Autophagy Machinery to Foster Replication

Macroautophagy, here referred to as autophagy, is a conserved endolysosomal cellular mechanism that coordinates the engulfment of cytoplasmic material into autophagosomes. Autophagosomes are fated to degradation and recycling after lysosomal fusion, forming the autolysosomes [26]. The acidic component of the lysosomes and autolysosomes is essential for the digestion of cargo originated from endocytosis, macropinocytosis, and autophagosomes. Thus, autophagy interacts with the endosomal pathway of the lysosomes [27] and plays critical roles in several physiological and pathological conditions, including cell survival and death, aging, metabolism, immunity and infection [28,29,30,31,32,33].

Autophagy is triggered by the inhibition of mammalian target of rapamycin complex 1 (mTORC1), the primary regulator of nutrient signaling. It has been demonstrated that mTORC1 integrates various stimuli and signaling networks to promote anabolic (e.g., protein synthesis) and inhibit catabolic (e.g., autophagy) processes [34]. Moreover, the mTORC1 complex is modulated by upstream regulators that transduce growth factors and energy signals. For example, stimulation of the phosphoinositide 3-kinase/serine-threonine kinase (PI3K/AKT) activates mTORC1 [35,36], while the adenosine monophosphate-activated protein kinase (AMPK), a sensor of cellular energy levels, inhibits the activity of mTORC1 [37]. Upon activation, mTORC1 inhibits downstream Netrin receptor (Uncoordinated protein 5) (UNC-5) effectors, such as the ULK1 (Unc-51 like autophagy activating kinase 1) complex, which phosphorylate the autophagy initiation machinery, including the autophagy-related protein 13 (Atg13) and ULK1 [38,39].

Autophagosome formation and its self-assembly are coordinated by enzymes and proteins located in the ER, such as phosphatidylinositol 3-phosphate (PI3P) and B-cell lymphoma 2 (BCL-2) interacting proteins Beclin-1/vacuolar protein sorting 34 (Beclin-1/Vps34) complex [40,41]. In addition to these proteins, the activating molecule in Beclin-1-regulated autophagy (Ambra1) plays a vital role as a regulator of autophagy, binding to Beclin-1, promoting the autophagosome formation [42]. The stabilization of the Beclin-1/Vps34 complex and initiating interaction among these autophagy regulators leads to the formation of PI3P and the recruitment of WD repeat domain phosphoinositide-interacting protein (WIPI) proteins to the autophagosome membrane. It has been reported that the conjugation of WIPI2b to PI3P promotes the lipidation of microtubule-associated proteins 1A/1B light chain 3A (LC3) and the elongation of the membrane [43]. In the autophagosome inner membrane, LC3-II, a mammalian homolog of Atg8 in yeast, plays a vital role in self-assembly, elongation and closure of the DMVs [44], which upon autophagy activation, a phosphatidylethanolamine is added to cytoplasmic LC3-I forming LC3-II, which translocate to the membrane [45]. LC3-II remains on the autophagosome membrane until fusion with acidic lysosomes, resulting in the formation of autolysosomes and completion of the degradation process (autophagic flux) [44]. While the fusion of the autophagosome with the lysosome is not fully understood, evidence suggests that specific membrane fusion proteins, called N-ethylmaleimide-sensitive factor attachment protein receptor (SNARE) complexes, participate in this process. Additionally, autophagosome-lysosome fusion is mediated by the interaction of Syntaxin 17 (Stx17), present in the external membrane of the autophagosome, with synaptosome-associated protein 29 (SNAP29) and vesicle-associated membrane protein 8 (VAMP8), both localized to the lysosome, consequently promoting the membrane fusion and subsequent cargo degradation [46].

Components of the autophagy machinery also participate in the secretion of invading pathogens. For example, poliovirus uses LC3-positive DMVs to escape the host cell defenses via a secreted autophagosome coated with the cellular components of the host, enabling a non-lytic release of virions [47,48]. Indeed, multiple viruses have evolved strategies to avoid host virophagic responses, including the expression of Beclin-1 inhibitors [49,50], proteins that inhibit the fusion of autophagosomes with lysosomes [51,52] and miRNA targeting autophagy and type 1 interferon signaling [53].

The use of autophagic machinery by CoVs was demonstrated, where the initiation of vesicle formation was inhibited by knocking out autophagy-related gene 5 (*ATG5*) or by wortmannin, suggesting that nsp6-induced autophagy was dependent on Atg5 and PI3K. Finally, transfecting the SARS-CoV open reading frame -8b and -3a into 293T and HeLa cells triggers lysosomal damage and ER stress, consequently inducing the translocation of Transcription Factor EB (TFEB) to the nucleus, a master regulator of lysosomal biogenesis and favoring the transcription of autophagy- and lysosome-related genes [54,55].

Defects in the molecular machinery for macroautophagy, such as the genetic inhibition of *ATG5* or beclin-1 (*BECN1*) genes, consequently make mice and primary human astrocytes more susceptible to viral infections [56,57,58]. Efficient virophagic responses also involve p62 and Atg5 [56]. Conversely, human immunodeficiency virus (HIV)-1^+^ patients who remained clinically stable for years in the absence of therapy display higher amounts of autophagic vesicles and high expression of autophagic markers in the peripheral blood mononuclear cells [59].

Conversely, other studies have highlighted the inhibitory effects of CoV nonstructural proteins on autophagy flux. In fact, overexpressing CoVs membrane-associated papain-like protease PLP2 (PLP2-TM) resulted in inhibition of autophagosome–lysosome fusion and blockade of autophagic flux in HEK293T, HeLa and MCF-7 cells [52]. Likewise, recent evidence described that Vero B4 cells infected with MERS-CoV exhibited reduced Beclin-1 levels, enhanced K48-polyubiquitylation of Beclin-1, reduced Atg14 oligomerization and blocked autophagosome-lysosome fusion [60]. Correspondingly, temporal kinome analysis of Huh7 and MRC5 cells infected with MERS-CoV displayed upregulated PI3K/AKT/mTOR and extracellular signal-regulated kinase/mitogen-activated protein kinase (ERK/MAPK)-mediated signaling [61]. Moreover, the double-stranded DNA human papilloma virus (HPV) promotes the inhibition of autophagy in infected host cells [62]. Mechanistically, HPV invasion and infection cause the stimulation of PI3K/AKT/mTOR pathway, increasing protein synthesis and reducing autophagy [63]. Additionally, HPV infection reduces many genes related to autophagy, including beclin-1 gene (*BECN1*) [64]. On the other hand, RNA viruses such as influenza A [65], porcine parvovirus [66], enterovirus A71 [67], dengue virus [68] and Zika virus [69], induce autophagy to help with their replication, and avoid recognition and degradation through autophagy. The mechanism includes inhibition of translocation of Stx17 and SNARE proteins, compromising autophagosome–lysosome fusion, and proteases that cleaves sequestosome-1 (SQSTM1) to prevent detection of viral target [70]. Overall, several studies have reported that the regulation of autophagy upon viral infection depends on factors related to the virus and the host cell, as also reviewed by Chiramel et al., (2013) [71].

An in-depth analysis of autophagy signaling, and metabolomics corroborated the notion that CoVs modulate PI3K/AKT/mTOR and AMPK signaling, showing that SARS-CoV-2 reduced glycolysis and protein translation by limiting the activation of mTORC1 and AMPK. It was shown that SARS-CoV-2 infection also downregulated spermidine and facilitated AKT1/S-phase kinase-associated protein 2 (SKP2)-dependent degradation of Beclin-1 [72]. Additionally, in particular, viral xenophagy (virophagy), an autophagic response directed to fully formed cytoplasmic virions or viral components [73], helps direct the virus to degradation, by presenting antigens and recognizing viruses, thus regulating inflammation and releasing proinflammatory cytokines [74]. Finally, some studies challenge the notion that virus replication depends on the autophagy proteins of the host. In fact, it was previously shown that *ATG5* or *ATG7* are not required for mouse hepatitis virus [75,76] or SARS-CoVs [77] replication, since knocking out of these genes did not inhibit viral infection.

Since autophagy may be one of the molecular mechanisms that allow cell invasion and virus replication, it is possible that some mutations may alter the autophagic process [78,79]. SARS-CoV-2, like any type of virus, accumulates mutations over time, and most of these mutations do not implicate in biological effects. However, some key mutations can alter viral biology to the extent of causing changes in its transmission and infection capacity [80]. Although SARS-CoV-2 does not have a high mutation rate (less than 25 predicted mutations per year), effectively identifying and tracking these mutations is of paramount importance for defining epidemiological parameters, and monitoring the evolution of the pandemic [80,81]. To this end, the genomic surveillance is currently carried out using several tools to identify these mutations, such as genomic and interatomic analysis web tools, phylogenetic analysis and the use of the network-based genetic divergence studies, which provide information to identify possible characteristics related to drug resistance and vaccine evasion [80,82,83].

Benvenuto et al., (2020) described mutations that affect the non-structural protein 6 (Nsp6), a protein encoded by the CoVs genome that binds to the host’s ER, promoting the generation of autophagosomes [79,84]. These mutations, in theory, should favor the affinity of the nsp6 protein with the ER, allowing a more stable binding between these components [79]. It is known that this binding favors viral infection, since it compromises the function of autophagosomes to deliver viral components for degradation in the host’s lysosomes [79,84]. However, the authors emphasize the need for more studies that can prove this interaction [79]. Additionally, in December of 2020, a preprint was published stating that one of the most relevant mutations in SARS-CoV-2, described as D614G, can increase the lysosomal traffic of the virus spike in infected cells, accelerating its process of entry into uninfected cells. The authors postulate that this is a possible explanation for the higher rates of transmission and infectivity promoted by this mutation [85]. New strains and clades of SARS-CoV-2 will emerge over time and some might eventually be resistant to drugs and vaccines that are currently being used or developed. However, currently available data demonstrate that the identified mutations so far did not alter the structure of the virus enough to create mechanisms to evade existing vaccines [86,87].

More recently, interactome studies revealed extra or intracellular interactions between SARS-CoV-2 proteins, not only Spike, and the host cell, involving endocytosis, autophagy and signaling. In a SARS-CoV-2 RNA interactome study from Huh7 cells, researchers observed 12 translation factors 24 h after infection [88]. Among them, components eukaryotic translation initiation factor Gamma 1 and 4B, respectively, EIF4G1, EIF4B, from the components eukaryotic translation initiation factor 4F (EIF4F), which is known to control mTOR activity [88]. This study also showed several interactors associated to vesicle trafficking proteins, such as SCFD1, USO1, RAB1A, RAB6D, RAB6A, RAB7A and GDI2 [88]. Additionally, Gordon et al., (2020) observed 332 protein interactions between SARS-CoV-2 and human proteins, among them, also proteins associated to vesicle traffic, such as Nsp6, Nsp7, Nsp10, Nsp13, Nsp15, open-reading frame 3a (Orf3a) and 8 (Orf8) [89].

Furthermore, Kliche et al., (2021) observed that the LC3-interacting region (LIR) in integrin β3 binding to the Atg8 domains of the autophagy receptors microtubule-associated protein 1A/1B-Light Chain 3 (MAP1LC3) MAP1LC3 and gamma-aminobutyric acid receptor-associated protein (GABARAP) in an enhanced manner by LIR-adjacent phosphorylation [90]. However, LC3 or green fluorescent protein (GFP)-LC3 failed to colocalize with viral replication-transcription complexes in SARS-CoV-infected Vero cells [91]. Thus, it appears as though non-canonical autophagy and/or unique components of the autophagy machinery are sequestered regardless of their activity in autophagic processing, which would mediate the induction of autophagy during CoVs infection [92]. These findings strongly support the idea that CoVs bypass autophagy to promote their replication. In fact, activation of autophagy and the endocytic pathway seems to play an important role in cell invasion and viral replication of CoVs, providing strong evidences for pharmacological targets development. However, gaps in knowledge are yet to be settled, as pointed and previously reviewed by Yang and Shen (2020) [78].

Briefly, Table 1 summarizes the molecular machinery recruited in autophagy initiation and Figure 1 shows that autophagy mechanisms represent potential targets for pharmacological inhibition of CoVs infection and replication.

## 4. Autophagy-Related Therapeutic Targets for COVID-19 Management

The network of endosomal–autophagic vesicles appears to play a central role in CoVs infection, including SARS-CoVs [60,78,94]. It is known that autophagy plays a role in pulmonary infections, enhancing the immune defenses of host against viral and bacterial infections of respiratory tract [95,96]. Considering the different roles autophagy plays during viral infection [97,98,99], we postulate that three groups of autophagy modulators could inhibit viral replication and are clinically relevant to COVID-19. The first group consists of drugs with lysosomotropic properties, which inhibits cathepsin activity and could prevent CoVs infection by neutralizing the acidic pH of the endosomes–lysosomes [100]. The second group is composed of protease inhibitors, which could inhibit the proteolytic cleavage of the S protein and consequently restrict viral cell entry [101]. The third group contains PI3K/AKT/mTOR regulators that, although considered autophagy regulators, could prevent the CoVs-mediated appropriation of the autophagic machinery [102,103]. In the following subsections, we will discuss several clinically approved and well-tolerated autophagy-modulating compounds that could be explored as potential modulators of SARS-CoV-2 infection and replication for the management of COVID-19.

### 4.1. Lysosomotropic Agents

#### 4.1.1. Chloroquine and Hydroxychloroquine

Chloroquine (CQ) and hydroxychloroquine (HCQ) are weak diprotic bases used as antimalarial drugs (Figure 2). These compounds accumulate in the endosome–lysosomal network of cells and neutralize the acidic pH, with consequent blockage of cathepsin activity and lysosomal fusion [104,105]. Previous studies showed that CQ displays a wide-spectrum of antiviral effects against CoVs, chronic HIV, and influenza viruses type A and B, both in vitro and in vivo [106,107].

Recent in vitro evidence, employing kidney-derived Vero cells, has indicated that CQ/HCQ could be effective at controlling the COVID-19 pandemic [108]. Indeed, several clinical trials investigating the efficacy of CQ/HCQ were initiated [109]. However, it should be pointed out that a recent study reported that the engineered expression of TMPRSS2 in Vero cells attenuates CQ-mediated antiviral activity, suggesting that SARS-CoV-2 infection may occur through multiple mechanisms [110].

Furthermore, a recent review on the use of CQ/HCQ for the treatment of COVID-19 with a small number of patients showed positive results in the recovery of infected patients. However, data from a study with a larger number of patients did not reveal any significant improvement in the symptoms of the disease. Instead, they highlighted the potential hazards to the health of the patients due to dangerous side effects [111], including retinopathy and increased waves QT interval in the electrocardiogram [112].

Additionally, on 15 June 2020, the FDA revoked the use of HCQ/CQ for COVID-19 treatment [113]. Then on 17 June 2020, the WHO announced that the HCQ arm of the solidarity trial for potential COVID-19 therapies would be discontinued [114]. Both of these actions were in response to the Randomised Evaluation of COVID-19 Therapy (RECOVERY) trial report conducted by Oxford University on patients from National Health Service (NHS) hospitals in the United Kingdom published on 5 June 2020. The chief investigators stated that a randomized trial, with a total of 1542 patients treated with HCQ compared to 3132 patients under usual care, found no significant differences in the mortality endpoint (25.7% HCQ vs. 23.5% usual care), hospital stay duration or any other outcome [115].

The FDA memorandum also reviewed other randomized, open-label and retrospective studies and highlighted that the results regarding differences in viral RNA shedding, when comparing HCQ/CQ-treated patients with others who did not receive these medications, were inconsistent. Thus, the FDA concluded that it is no longer reasonable to believe that the oral formulations of HCQ/CQ were effective at treating COVID-19 [113].

Nonetheless, it is plausible that the mechanistic insights related to the CQ/HCQ mode of action could lead to the development of safer and more effective COVID-19 therapies [100]. For example, in vitro studies showed that CQ induced intracellular retention of ACE2 and abrogated the SARS-CoV-2-receptor binding at the cell surface and cellular entry [116]. Similarly, cell cultures treated with CQ, NH_4_Cl or bafilomycin A1 (an endo/lysosomal V-ATPase inhibitor) resulted in ACE2 receptor arrest within the perinuclear vacuoles, suggesting that lysosomotropic agents may share pharmacological targets [117]. Nonetheless, the lysosomotropic agents could interfere in the ACE2 action, avoiding the viral invasion and entry in host cells, since this entry route constitute an important pathway for viral invasion and cell infection [78]. In fact, recent studies have demonstrated that drugs with mechanistic similarity to CQ/HQC revealed inhibitory activity against SARS-CoV-2. For instance, GNS561, a small basic lipophilic molecule that induces lysosomal dysregulation and inhibition of the late-stage of autophagy, demonstrated potent in vitro antiviral activity against the novel SARS-CoV-2 alone and in combination with remdesvir [118]. Currently, GNS561 is being tested in cancer patients with moderate COVID-19 (National Institute of Health (NIH)-Clinical Trials Database; Identifier: NCT04333914) [119]. A recent study found that four lysosomotropic autophagy-inhibiting compounds—namely ROC-325, mefloquine, hycanthone and clomipramine—blocked the cytopathic effect of SARS-CoV-2 in Vero-E6 cells [120]. Moreover, the cytopathic effect against the SARS-CoV-2 correlated with LC3 puncta in antiviral doses [120]. Nonetheless, taken together, the current evidence discards HCQ/CQ as a repurposed treatment with potential to manage COVID-19 [115].

#### 4.1.2. Azithromycin

Azithromycin is a broad-spectrum macrolide antibiotic that binds to the S50 ribosomal subunit of bacteria inhibiting its protein synthesis [121] (Figure 2). The antiviral efficacy of azithromycin has been demonstrated in different viral infections [122,123,124,125]. For example, in 2019, an in vitro study evaluated azithromycin against influenza A (H1N1) infection and found that it blocks the cellular internalization of the virus and inactivates its endocytic activity [125]. More recently, a study using respiratory epithelial cells of cystic fibrosis showed that azithromycin has an antiviral action similar to HCQ, acting as a weak acidophilic lipophilic base and increasing the pH of organelles such as the endosome and the trans-Golgi [126].

Interestingly, other studies have reported synergism between HCQ and azithromycin in the treatment of SARS-CoV-2 infections [127,128]. This recommendation is based on recent in vitro data showing that azithromycin and CQ cotreatment reduces SARS-CoV-2 replication [127]. Notably, clinical trials with this drug combination have been conducted [129,130,131,132], but the available data are divergent and inconclusive [133,134]. Of note, the conclusions made by Gautret et al., (2020) have been considered by Nguyen et al. (2021) as not based on a rigorous study design or analysis, being the effect of HCQ and azithromycin on antiviral effect remaining uncertain. Additionally, there are risks of secondary effects of this treatment, such as heart complications [130,132]. Thus, for now, there is no significant benefit of azithromycin, or its association with CQ/HCQ, for the treatment of COVID-19.

#### 4.1.3. Artemisinin and Its Derivative Compounds

Artemisinin is isolated from the herb *Artemisia annua* L. [135]. The derivative compounds are sesquiterpene lactones with a unique endoperoxide bridge moiety primarily responsible for their biological actions [136] (Figure 2). The compounds target several signaling pathways involved in the inhibition of autophagy, due the lysosomal disruption, release of cathepsins and other hydrolytic enzymes into the cytosol and subsequent cell death [137].

Studies have also demonstrated that, in cancer cells, artemisinin and its derivatives regulate proteins involved in several autophagy signaling pathways. Guan and Guan (2020) showed autophagosomes visualization, a hallmark of autophagy, in cancer cells exposed to artemisinin [138]. Additionally, these compounds inhibit mTOR, nuclear factor kappa B (NF-κB), PI3K and the signal transducer and activator of transcription 3 (STAT3) [136].

Traditionally, artemisinin was used for fever treatment and recently it has been approved as a therapy for malaria [139]. Previous data showed that artemisinin complexes are effective against viral infections, such as human cytomegalovirus (HCMV), human herpes simplex virus 1 and 2 (HSV-1 and HSV-2), hepatitis C virus (HCV), hepatitis B virus (HBV), HPV 39 and polyomavirus BK [140]. Moreover, artemisinin modulates inflammatory and immunologic responses in doses that are relatively safe, with a low toxicity profile [141]. Concerning the anti-inflammatory properties, artemisinin regulates the expression of pro and anti-inflammatory cytokines (e.g., IL-1β, IL-2, IL-6, IL-8 and IFN-γ) and NF-κB, and has been used to treat different respiratory diseases [142]. The regulation of NF-κB expression is of particular importance since its inhibition decreased the severity of the acute respiratory syndrome and increased the survival of SARS-CoV infected mice [143]. These data suggest that artemisinin and related compounds may show interesting results against SARS-CoV-2 infections.

At the moment, ArtemiC, a micellar formulation of artemisinin, curcumin, frankincense (*Boswellia*) and vitamin C, which is administered by spraying, is in phase II of clinical trials for patients diagnosed with COVID-19 (NIH-Clinical Trials Database; Identifier: NCT04382040 and NCT04553705) [119]. Additionally, another phase II clinical trial investigates the effects of artemisinin/artesunate on the course of the disease and viral load is currently underway in patients with COVID-19 (NIH-Clinical Trials Database; Identifier: NCT04387240) [119]. So far, no clinical evidence is available as none of the studies are yet concluded, but a very recent preprint study demonstrated anti-SARS-CoV-2 activity in vitro of three artemisinin compounds: artesunate, arteannuin B and lumefantrine [144].

#### 4.1.4. Two-Pore Channels Modulating Agents

Nicotinic acid adenine dinucleotide phosphate (NAADP) is an intracellular messenger that plays a vital role in the mobilization of Ca^2+^ in mammalians cells [145,146,147,148] by binding to two-pore channels (TPCs) [149,150]. Furthermore, NAADP has been reported as a potent Ca^2+^ mobilizing messenger and inducer of autophagy [151,152,153,154,155,156]. On the other hand, the TPC antagonists ned-19 and tetrandine (Figure 2) were postulated as possible blockers of lysosomal function, causing a further inhibition of autophagy on the degradation step [152].

Interestingly, TPCs have been identified as a new host factor for Ebola virus (EBOV) entry, and their inhibition prevents EBOV infection [157,158]. Similarly to CoVs, EBOV enters the host cell and moves through the endolysosomal system, using the cell machinery of the host for its replication and releasing its genome into the cytoplasm [159]. Moreover, TPC1 or TPC2 colocalize with MERS-CoV S protein in Huh7 cells, providing evidence that TPCs may also regulate MERS-CoV viral entry [160]. In this study, the authors observed that several Na^+^ channel blockers and voltage-operated Ca^2+^ antagonist drugs, including ned-19 and tetrandrine, attenuated MERS-CoV translocation in Huh7 cells. Moreover, the complete inhibition of Ca^2+^ release was not related to lysosomotropism, and no alterations in lysosomal pH were detected after applying these drugs. Thus, based on the fact that stimulation of the transient receptor potential mucolipin 1 (TRPML1) failed to control MERS-CoV pseudovirus infectivity, it was concluded that the TPC function might be required for this infection [160]. These findings suggest that TPC activity is essential for the passage of EBOV and MERS-CoV virus through the endolysosomal trafficking pathway, and that Ca^2+^ channel ligands might hinder viral infectivity. Currently, the only clinical trial employing tetrandrine (approved for Clinical Trials for COVID-19, NIH-Clinical Trials Database, 2020; Identifier: NCT04308317) [119] is still recruiting and not yet concluded, refraining any conclusion about clinical efficacy. Thus, understanding the role that TPCs play in viral infectivity could lead to the discovery of novel antiviral agents, maybe against SARS-CoV-2.

#### 4.1.5. Imatinib

Imatinib, a tyrosine kinase inhibitor developed in 2001, revolutionized the treatment of chronic myeloid leukemia [161], since its activity against the breakpoint cluster region gene-Abelson proto-oncogene (BCR-ABL) in cancerous cells [162] (Figure 2).

Several ongoing studies are exploring imatinib in other pathologies that are associated with its target kinases. For example, García et al., (2012) proposed that imatinib could inhibit EBOV replication and release by blocking c-ABL1 and VP40 phosphorylation [163]. In addition, preclinical data demonstrated that imatinib inhibits the fusion MERS-CoV and impairs endosomal trafficking in vitro [164]. Since the ABL-2 activity is essential for sequential steps involving fusion to viral replication, its inhibition by imatinib leads to impaired replication of SARS-CoV and MERS-CoV in vitro [162,165].

Concerning COVID-19, Morales-Ortega et al., (2020) administered imatinib (400 mg/day) to a SARS-CoV-2 infected patient who was progressing to a severe inflammatory state after 12 days of symptoms [166]. After three days of treatment with imatinib, the fever disappeared, oxygen supplementation was interrupted and the radiological stability of pulmonary opacities was confirmed. There was also an improvement in laboratory parameters after the 5th day of treatment [166]. Currently, there are four ongoing clinical studies (phase II/III) with imatinib being conducted in accordance with the NIH (NCT04394416, NCT04422678, NCT04346147 and NCT04357613). Such clinical trials are testing the efficacy of this drug alone, or in combination with other drugs, in the treatment of COVID-19 (NIH-Clinical Trials Database) [119]. So far, no clinical evidence is available as studies are under recruitment stage and not yet concluded. The results of these studies will provide a scientific basis for this pharmacological application.

Taken together, CQ/HCQ alone or in combinations with macrolides failed as an antiviral, immunomodulatory or prophylactic therapy against COVID-19 [115], but other lysosomotropic agents, such as imatinib, remains as promising strategies [167,168,169]. Sauvat et al., (2020) discuss SARS-CoV-2-inhibiting lysosomotropic agents as “on-target” versus “off-target” and presents a rationale for their clinical application. Briefly, “off-target” agents display unspecific action upon acidophilic organelles including autophagosomes, endosomes and lysosomes, with CQ/HCQ as the prototypical example for the class. The non-specific effects of “off-target” agents result in amplified safety issues related to side effects versus antiviral activity ratio. Thus, clinical trials conducted with these drugs are expected to fail due to low efficacy and safety [170]. In contrast, “on-target” agents mediate their effects through specific mechanism, which results in reduced side-effects and increased therapeutic index. For instance, imatinib has demonstrated few side effects in the long-term treatment of chronic myeloid leukemia [171] and gastrointestinal stromal tumors [172]. Thus, the pursuit for lysosomotropic agents should focus on agents with a well-defined therapeutic target in order to enhance a therapeutic index and raise drug efficacy and safety against COVID-19.

### 4.2. Protease Inhibitors/Antiviral Agents: The Prevention of Infection

#### 4.2.1. Camostat Mesylate

Camostat mesylate inhibits the serine protease TMPRSS2 and prevents the entry of SARS-CoV-2 into the host cells [101] (Figure 2). Other proteases, including cathepsin-L, thermolysins, plasmins and trypsin, can act as a cofactor for virus entry into the host cell [173]. In this sense, strategies that target the inhibition of TMPRSS2 activity could block SARS-CoV-2 cell invasion and potentially hinder autophagy machinery appropriation by SARS-CoV-2 virions [174]. Indeed, in vitro and in vivo studies have described the inhibitory effect of camostat mesylate on SARS-CoV invasion and infection.

In mice, Zhou et al., (2015) demonstrated that camostat mesylate was effective against SARS-CoV infection, reducing the pathogenesis and increasing the survival rate of animals exposed to the virus. The same authors also suggested that similar results might be observed with MERS-CoV infections [175]. Furthermore, in vitro studies performed by Hoffmann et al., (2020) showed that camostat mesylate-mediated inhibition of TMPRSS2 reduced SARS-CoV-2 entry and infection of the human lung Calu-3 cell line [101]. According to the NIH (NIH-Clinical Trials Database) [119], fourteen clinical trials are recruiting and will investigate the efficacy of camostat mesylate, alone or in combination with other drugs, in patients with COVID-19. One of these studies was withdrawn due to lack of public funding and evidence as the planned control arm with HCQ treatment showed out as not being standard of care anymore as time evolved (NCT04338906).

#### 4.2.2. Lopinavir

Lopinavir (ABT-378) is a potent protease inhibitor used to prevent HIV replication and spread [174] (Figure 2). It has been suggested that since SARS-CoV-2 contains structural components that are similar to other viruses, including HIV, it is plausible that this antiviral therapy could be used to treat patients with COVID-19 [176].

The administration of lopinavir/ritonavir to marmosets infected with MERS-CoV demonstrated that it could reduce the disease progression and improved clinical outcomes [177]. In a randomized, controlled, open-label trial conducted in Wuhan (China), 199 patients with COVID-19 were treated with lopinavir/ritonavir (400 mg/100 mg) twice a day for 14 days in combination with standard care treatment or standard care treatment alone [178]. Results showed that drug-treated patients did not demonstrate clinical improvement or reduced mortality after 14 days. Additionally, the drug combination failed to attenuate the viral RNA load, which was assessed in patients at the end of the trial.

At the moment, more than eighty clinical trials are registered on the NIH website with lopinavir/ritonavir or in combination with ribavirin or interferon β1a are in the initial phases or ended (NIH-Clinical Trials Database; Identifier: NCT04276688) [119]. Results of the concluded studies are not available yet, restraining any clinical conclusion, while withdrawn studies occurred only due to epidemiological dynamics and lack of funding.

#### 4.2.3. Umifenovir

Umifenovir is currently used in Russia and China as a prophylaxis for the treatment of pulmonary infections caused by human influenza A and B viruses and HCV [179,180] (Figure 2). The proposed mode of action of umifenovir involves intercalation with membrane lipids, inhibiting viral fusion with the plasma membrane of the host cell. It has also been shown that the drug can bind to the membrane-bound clathrin protein and prevent endocytosis of the virus [179]. It has been suggested that umifenovir may be effective against EBOV and Lassa virus, highly pathogenic agents that caused outbreaks in the West African region [181,182]. Additionally, in vitro studies demonstrated that umifenovir displays antiviral activity against the SARS-CoV viruses [183].

Considering that umifenovir binds directly to membrane phospholipids and endosomal vesicles, it could interfere with the autophagic flux of the host cell. However, there have been no studies linking the action of the drug with autophagy. Based on these results discussed above, umifenovir is a promising drug against CoVs. Currently, ongoing randomized clinical trials evaluating the efficacy and safety of umifenovir against COVID-19 are being conducted in China, Turkey and Iran (NIH-Clinical Trials Database, 2020; Identifier: NCT04350684) [119], but they are not yet concluded or do not have available results.

#### 4.2.4. Teicoplanin and Others

Teicoplanin is a clinically approved glycopeptide antibiotic that inhibits cathepsin L activity and blocks MERS-CoV and SARS-CoV entry into cells [184] (Figure 2). More recently, this drug also showed antiviral activity against SARS-CoV-2 [185]. Similarly, another cathepsin L inhibitor, Z-FY(t-Bu)-DMK, and cysteine protease inhibitors E64d and K11777, have been shown to block the SARS-CoV infection [103,186]. Moreover, MG132, a proteasome and cysteine protease inhibitor, and MDL28170, an m-calpain inhibitor, effectively inhibited SARS-CoV replication [103]. These studies are based on the fact that SARS-CoV entry requires cathepsin L, cysteine protease and serine protease activity, and thus define viable pharmacological targets for COVID-19 management.

Taken together, antiviral agents proposed for SARS-CoV-2 infection treatment focus on preventing host cell invasion. For that reason, they act as a barrier against SARS-CoV-2 infection. Camostat mesylate, lopinavir and teicoplanin are potent protease inhibitors capable of hindering the S-protein cleavage required for viral infection. Umifenovir, on the other hand, does not display protease inhibitory activity, but hampers viral fusion with host cell membrane resulting in the same desired effect as the protease inhibitors. As proposed in many clinical trials, the combination of these drugs with lysosomotropic agents or PI3K/AKT/mTOR modulators may result in additive or synergistic effects upon viral replication, targeting multiple mechanisms involved in viral infectivity.

### 4.3. PI3K/AKT/mTOR Modulators

#### 4.3.1. Rapamycin

Rapamycin is a PI3K/AKT/mTOR inhibitor and clinically proven macrolide that exhibits potent antitumor and immunosuppressive activity [103,186] (Figure 2). While the antiviral activity of rapamycin is controversial [103], it was capable of reducing porcine epidemic diarrhea virus [187], transmissible gastroenteritis virus (TGEV) and CoVs infectivity [188].

Regarding specifically to CoVs, the PI3K inhibitor wortmannin inhibited MERS-CoV infection in Huh7 cells [61] and reduced vesicle formation in HEK cells that express infectious bronchitis virus (IBV) nsp6, thus indicating that nsp6-induced autophagy was dependent on PI3K [84]. Likewise, the inhibition of PI3K with VPS34-IN1 in Vero E6 cells and its bioavailable analogue VPS34-IN1 in ex vivo human lung tissues potently suppressed SARS-CoV-2 replication at a nanomolar level [189]. Moreover, the pharmacological inhibition of E3 ubiquitin ligase, a component of SKP2, decreased the ubiquitination and degradation of Beclin-1 and enhanced autophagic flux, consequently reducing MERS-CoV replication [60].

A very recent preprint reported that SARS-CoV-2 infection limits autophagy by interfering with various metabolic pathways and that compound-driven interventions aimed at inducing autophagy reduced the spread of SARS-CoV-2 in vitro. It has also shown that spermidine, MK-2206 (an AKT inhibitor), and niclosamide (a Beclin-1 stabilizing anthelminthic drug) inhibited the in vitro spread of SARS-CoV-2 by targeting these pathways [72].

In fact, upregulation of the PI3K/AKT/mTOR signaling pathway occurs during SARS-CoV-2 infection, as revealed by proteomics and transcriptomics data [190]. Authors showed the activation of AKT/mTOR signaling during initial phases of infection, and the inhibition of AKT by MK-2206 can suppress SARS-CoV-2 infection. Nonetheless, rapamycin has not shown an effective action to limit the infection. As mentioned above, Kindrachuk et al., (2015) showed that upregulation of the PI3K/AKT/mTOR pathway also occurs in MERS infection, which suggests its pivotal role in CoVs infection [61]. Thus, PI3K/AKT/mTOR inhibitors are attainable strategies to alleviate CoVs infections, such as MERS-CoV and SARS-CoV. Once the inhibition of this pathway with a library of kinase inhibitors suppresses the MERS infection, the use of PI3K/AKT/mTOR inhibitors represent a novel strategy to prevent SARS-CoV-2 infection.

In this way, not only the activation of autophagy with mTORC1 inhibitors could play a role in the SARS-CoV-2 infection, but compounds that can act in the PI3K/AKT/mTOR pathway, like the MK-2206, can promote the suppression of viral replication and spread, once this pathway regulates many cellular processes and not only autophagy. Until now, clinical trials employing rapamycin or other mTOR inhibitors are not concluded and are still recruiting patients (NIH-Clinical Trials Database; Identifier: NCT04341675, NCT04461340 and NCT04584710) [119], hindering any conclusion about clinical efficacy. Nevertheless, one study was withdrawn due to irregular admission to hospital and shifted approaches from repurposing old drugs (NCT04371640).

#### 4.3.2. Heparin

Heparin exhibited several antiviral actions [191,192,193,194], probably due to its structural similarity to heparan sulfate [195], a glycosaminoglycan formed by proteoglycans present on the surface of cells that participates in viral entry into eukaryotic cells as an initial anchoring domain [191,196] (Figure 2). Thus, heparan sulfate appears to modulate the entry of SARS-CoV into cells. It has been shown that the SARS-CoV-2 Spike S1 receptor can bind to heparin, changing the receptor conformation [197]. Additionally, the treatment of Vero cells with heparin inhibited SARS-CoV-2 infection [197]. Additionally, using a heparin-like polysaccharide in cells infected with HPV, Gao et al., (2018) showed that sulfated chitasone attenuated the HPV infection in different cell lines and inhibited the PI3K/AKT/mTOR pathway, which is indicative of autophagy activation. These data suggest a relationship between heparin-related compounds with autophagy during viral infections. Concluded clinical trials employing heparin have not posted results so far (NIH-Clinical Trials Database; Identifier: NCT04359212 and NCT04518735) [119].

#### 4.3.3. Glucocorticoids

Glucocorticoids (GCs) are steroid hormones with potent anti-inflammatory and immunosuppressive actions used in the treatment of chronic inflammatory, autoimmune and allergic diseases [198,199] (Figure 2).

During the SARS-CoV and MERS-CoV epidemics, GCs were widely used to decrease the exacerbated immune response caused by the uncontrolled release of proinflammatory cytokines observed during severe lung inflammation [200,201]. Consequently, GCs were proposed for the treatment of COVID-19 patients with mild to intermediate doses in an initial treatment of cytokine storm and in specific cases of COVID-19-induced pneumonia [202,203]. However, GCs may increase the risk of secondary infections and delay the clearance of the virus, as was observed with their use in infections caused by MERS-CoV and SARS-CoV [200,204].

In other viral infections, GCs exhibit an autophagy-dependent antiviral effect. For example, budesonide, a synthetic GC, was able to inhibit the replication of the HCoV-229E CoV and alters the luminal pH of acid endosomes [205]. Budesonide also reduced human rhinovirus (HRV) replication in HeLa cells by inducing autophagy, and this antiviral effect was attenuated in the presence of the autophagic blockers CQ and bafilomycin-A1 [206]. Similar effects were observed with dexamethasone in the same viral infection [207]. However, He et al. (2018) observed that dexamethasone stimulated HRV replication. The authors showed that this stimulation was autophagy-dependent since dexamethasone, in the presence of the autophagic inhibitor 3-methyl-adenine, reduced viral replication [208]. Together, these data show that the GC-mediated effect on viral replication depends on both the virus and the autophagy pathway.

In fact, GCs induce autophagy by negatively modulating mTORC1 [209,210,211,212] and the transcription of genes related to the mTORC1 pathways, such as MAPK/ERK and PI3K/AKT [213]. Additionally, GCs-induced autophagy involves the ubiquitin ligase TRIM32 (tripartite motif-containing 32) that is required for the induction of muscle autophagy under atrophic conditions [214].

Regarding SARS-CoV-2 replication, inhaled GCs reduce the expression of ACE2 and TMPRSS2 genes in patients with asthma [215] and attenuate ACE2 receptors in human and murine in vitro and in vivo models [216]. In agreement, steroidal sex hormones (estradiol, progesterone and testosterone) are implicated in the age-dependent and sex-specific severity of COVID-19 through mechanisms including modulation of the immune responses and ACE2 and/or TMPRSS2 levels [217,218,219,220]. For example, some evidence supports that estrogens and progesterone exert an immune-protective effect on women in COVID-19 by positively modulating immune T cells and a blockade of proinflammatory cytokine storm [221,222]. Estrogens can also downregulate ACE2 mRNA levels in bronchial epithelial cells in vitro [223]. In addition, 17β-estradiol treatment reduced the levels of the TMPRSS2, which are involved with SARS-CoV-2 infectiveness capacity, and, hence, also reduced SARS-CoV-2 viral load [224]. Both high and low testosterone levels can favor severe COVID-19 [225,226], as high testosterone levels may upregulate TMPRSS2, facilitating the entry of SARS-CoV-2 into host cells. It was observed that androgens, besides their immunosuppressive effects via inhibition of the proinflammatory cytokine storm [227], can strongly upregulate the expression of TMPRSS2 in prostate cancer cells and human lung epithelial cells [228]. Thus, preclinical data demonstrates that blocking the activity of TMPRSS2 protease through camostat mesylate, nafamostat or bromhexine decreases the entry of SARS-CoV-2 into lung cells and may improve COVID-19 infection in men [229,230]. After the RECOVERY trial report, GCs such as corticosterone, have been recommended by WHO for severely ill COVID-19 patients, comprising one of the few available therapy for COVID-19 management [231]. While the results with GC and hormone-therapy appear to be promising, further studies on the action of these drugs in SARS-CoV-2 infection are necessary.

#### 4.3.4. Angiotensin-Converting Enzyme Inhibitors (IECAs) and Type 1 Angiotensin II Receptors Blockers (ARB)

Several studies have shown that renin-angiotensin system (RAS) deregulation may be responsible for acute respiratory distress syndrome, which can be triggered by viruses (SARS-CoV, H5N1 and H7N9), bacteria and particles and molecules [232]. Therefore, excess angiotensin II may be primarily responsible for increased SARS-CoV pathogenesis [233]. Thus, these studies suggest that decreasing the angiotensin II levels or blocking the RAS pathway might attenuate acute lung injury severity. In the same context, a meta-analysis showed that the angiotensin receptor blocker and angiotensin-converting enzyme inhibitor (ACEI) reduce the risk of pneumonia and lower disease morbidity and mortality [234].

The type 1 angiotensin II receptor (AT_1_) also controls several physiological processes, including autophagy. H. Xu et al., (2020) found that mechanical stress triggers cardiomyocyte autophagy through AT_1_ receptors, activating p38MAP kinase-independent of angiotensin II [235]. The type 2 angiotensin II receptor (AT_2_) blocker PD1223319 failed to abolish autophagy, thus confirming that angiotensin II induces autophagy through AT_1_ receptors [236]. Furthermore, it was demonstrated that angiotensin II increases the number of autophagosomes in cells with high level of AT_1_ receptors and these effects were antagonized when cells coexpressed the AT_2_ receptor [237].

Additionally, the treatment of ACE inhibitors or ARB was effective on COVID-19 patients with lower complications. For instance, Zhang et al., (2019) found that ACE2 activation or inhibition in lung tissue affected the severity of acute lung injury by modulating levels of proinflammatory factors and autophagy induction through the AMPK/mTOR pathway [238]. In favor of these findings, hospitalized patients with COVID-19 using ACEI/ARB had a lower risk of disease-induced mortality when compared to non-users of these drugs [239]. Additionally, in a retrospective multicenter study conducted in China, with 1128 hypertensive patients diagnosed with COVID-19 (188 received ACE inhibitors or ARB and 940 without receiving ACEI/ARB), the mortality rate was higher in the population that did not receive ARB/ACEI drugs (9.8% vs. 3.7%) [239]. Nonetheless, results from completed clinical trials employing ACEI/ARB are not yet available (NIH—Clinical Trials Database; Identifier: NCT04318301, NCT04357535 and NCT04318418) [119].

#### 4.3.5. Cannabidiol

Cannabidiol, a phytocannabinoid from *Cannabis sativa*, is effective at treating arthritis, ear inflammation, inflammatory bowel disease, neuroinflammation and pulmonary inflammatory disease [240,241,242,243,244] (Figure 2).

Cannabidiol produces no psychotropic effects and has a safe and tolerable dose range, making it an attractive drug [245]. Aside from its anti-inflammatory and immunomodulatory effects, there is little if any evidence that cannabidiol could be effective against viral infections [246]. For example, cannabidiol decreased neuroinflammation by negatively regulating chemokine (C-C motif) ligand type 2 and 5 (CCL2 and CCL5) and the proinflammatory cytokine interleukin-1 β (IL-1β) induced by Theiler’s murine encephalomyelitis virus (TMEV) in mice [247]. These findings are supported by other research demonstrating the attenuated production and release of IL-1β, IL-6 and IL-β in BV-2 in lipopolysaccharide activated microglia treated with cannabidiol [248]. Cannabidiol treatment also decreased the levels of IL-4, IL-5, IL-13, IL-6 and tumor necrosis factor α (TNF-α) in an experimental model of asthma in rats, consequently reducing airway inflammation and fibrosis [249]. Along with these lines, it is known that SARS-CoV-2 infection leads to a proinflammatory cytokine storm [250]; thus, cannabidiol might decrease the levels of these cytokines and benefit patients infected with SARS-CoV-2 [251].

A group of researchers in Canada recently showed that extracts from *C. sativa* containing high levels of cannabidiol downregulate the expression of the ACE2 gene and TMPRSS2, which, as discussed previously, are the primary receptors for SARS-CoV-2 entry into host cells, in different models of human epithelia [252]. Additionally, cannabidiol can act as an antioxidant at several receptor sites, including the peroxisome proliferator-activated γ (PPARγ) and adenosine 2 receptors [253]. Concerning PPARγ, it is highly expressed in the alveolar macrophage in acute pneumonia and is responsible for controlling the pulmonary inflammatory processes that promote tissue recovery after viral respiratory infections [254]. In this sense, cannabidiol action at PPARγ receptor sites may produce a considerable improvement in lung function by preventing the cytokine storm of resident macrophages [243,254,255].

Cannabidiol induces autophagy in different cell types, which can play either a protective or harmful role, depending on the stimulus and exposure time [256]. For example, one study demonstrated that cannabidiol induced autophagy by increasing the formation of autophagosomes and inhibiting autophagosome degradation in an intestinal epithelium model [257]. The activation of autophagy also involves the ERK 1/2 and PI3K/AKT signaling pathways, which are modulated by cannabidiol [258]. A previous study reported autophagy activation via the ERK/MAPK cascade, leading to the attenuation of AKT phosphorylation induced by growth factors [259]. Furthermore, Hiebel et al., (2014) showed that autophagy could be modulated by the cannabinoid receptor type 1 (CB1) independently of the mTOR and Beclin-1 complex [260].

Presently, clinical trials employing cannabidiol are still recruiting and not concluded (NCT03944447). Thus, although further investigations are necessary to evaluate the effects of cannabidiol on viral infections, inflammation, immune system control and autophagy, there is a plethora of data supporting the hypothesis that it may be a safe and useful adjuvant therapy for SARS-CoV-2.

The coronavirus family is known to avoid autophagy and escape endosomal degradation [261,262], but whether these viruses induce or arrest the autophagy machinery is unclear. Nevertheless, autophagy favors immunity in respiratory diseases [263,264] as it facilitates the selective disintegration of immunogenic components associated with viral particles, benefiting pattern recognition in innate immune response and antigen presentation in adaptive immunity [262]. Coronavirus also upregulates PI3K/AKT/mTOR signaling, and kinase inhibitors such as wortmannin (PI3K inhibitor), MK-2206 (AKT inhibitor) and rapamycin (mTOR inhibitor) restrains CoVs infection in vitro [60,61,190]. The use of PI3K/AKT/mTOR inhibitors could hinder autophagy appropriation by CoVs, but also favor immunity and antigen presentation, and benefit the secretion of anti-inflammatory cytokines and tissue repair [95,265,266,267].

In order to offer an overview on the proposed mechanisms, we summarized several clinically approved and tolerated autophagy-modulating drugs described here with their respective ongoing clinical trials for the management of COVID-19 (Table 2), and their conventional therapeutic use and toxicological properties (Table 3). The results of these trials will be essential for a better evaluation of the clinical potential and evaluation of the therapeutic strategy, dose and posology, as a safe estimation of risk/benefit is very challenging without population parameters.

## 5. Conclusions

Due to its rapid spread, high lethality and impact on health systems, the COVID-19 pandemic caused by SARS-CoV-2 represents one of the most significant challenges ever faced by the modern world. The lack of knowledge about the virus and access to a viable vaccine has forced researchers and medical professionals to identify alternative compounds and drugs that can be effective in containing the pandemic. In this review, we discussed the potential of autophagy inhibitors in the treatment of COVID-19 infection, and offered a justification for the mechanism related to autophagy for the potential antiviral activity of these drugs.

In this regard, the initial debate orbited around lysosomotropic agents, such as CQ/HCQ, as potential off-label drugs for the treatment of COVID-19. Unfortunately, clinical trials have failed to demonstrate any therapeutic benefit for CQ/HCQ, since the lysosomotropic agents have shown limited efficacy and safety due to the nonspecific action on acidophilic organelles. In contrast, agents that mediate their effects through specific mechanisms, such as protease inhibitors and antiviral drugs, have superior clinical potential due to their reduced side effects. Several of these drugs are still under clinical investigation and are expected to be well tolerated and to reduce the severe clinical outcomes of COVID-19. At the same time, PI3K/AKT/mTOR inhibitors, such as rapamycin, are potent anti-inflammatory and immunosuppressive agents that can offer therapeutic benefits against COVID-19. For example, GCs induce autophagy by negatively modulating mTORC1 and are one of the few therapies available for the management of COVID-19 recommended by WHO. Nonetheless, the use of these drugs appears to be limited to specific clinical circumstances since GCs are indicated for severely ill patients who suffer from cytokine storm and aggravated inflammation. Finally, combined therapy with more than one of the proposed drugs should not be disregarded, as complementary antiviral mechanisms can offer additive therapeutic effects with few side effects to patients with COVID-19.

However, in order to establish the appropriate therapeutic strategy and define the risk/benefit of the proposed drugs, the conclusion of the clinical trials is essential. We hope that the drugs listed here can demonstrate a beneficial effect against COVID-19 in clinical trials and then integrate future international protocols for the treatment of COVID-19.

## Figures and Tables

**Figure 1 ijms-22-04067-f001:**
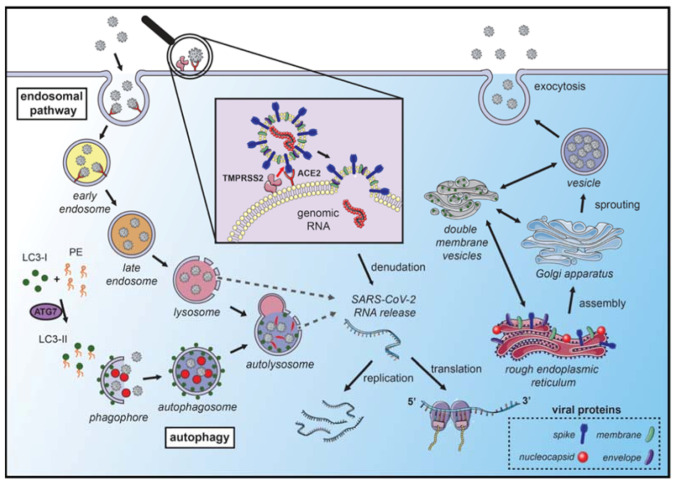
Coronavirus hijacks autophagy machinery to promote their replication. SARS-CoVs bind to the angiotensin-converting enzyme 2 (ACE2) receptor on the membrane surface and enter the host cell. The fusion with the membrane and the release of the genomic RNA into the cytoplasm occurs after the cleavage of the spike (S) protein, which can occur in several locations. S protein cleavage occurs on the cell membrane surface by the transmembrane protease serine 2 (TMPRSS2), which is associated with the ACE2 receptor, or by cathepsin-L and cysteine proteases in the endosomal system. The acidic pH in the lysosomes is necessary for the activity of cathepsin-L and S protein cleavage. Next, the endosomal cargo converges with the autophagic vacuoles in the lysosomes. Coronavirus nonstructural proteins colocalize with microtubule-associated proteins 1A/1B light chain 3A (LC3-II) in the endomembrane system, suggesting that autophagy plays a role in amplifying coronavirus replication. After fusion with the membrane, the genomic RNA is released and stripped of the nucleocapsid protein. Viral proteins are translated in the endoplasmic reticulum, which promotes the rearrangement of endoplasmic reticulum membranes and the formation of double-membrane vesicles, which are also localized with LC3 and autophagy-related proteins. The newly synthesized genomic RNA is then assembled into virions in intermediate compartments located between the endoplasmic reticulum and the Golgi apparatus and moves through the secretory pathway of the host and eventually released by exocytosis (the illustration was produced using the smart servier medical art vectors for publications and presentations licensed under the Creative Commons (CC BY 3.0)) [93].

**Figure 2 ijms-22-04067-f002:**
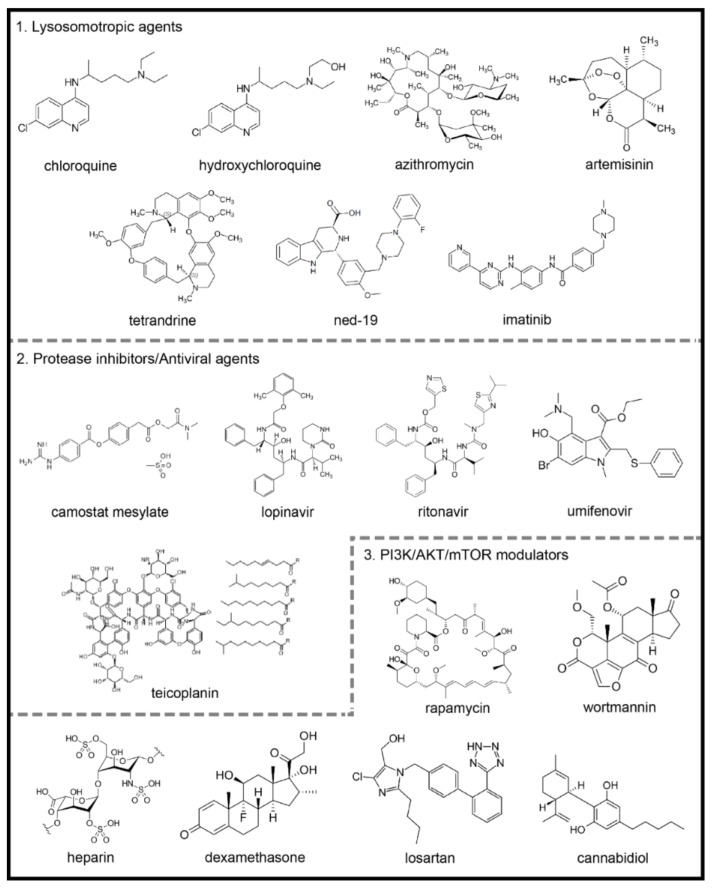
Chemical structures of potential autophagy-related drugs for SARS-CoV-2 infection. The drugs were divided in three groups according to their effects on the autophagy signaling pathway and possible effect against SARS-CoV-2 infection. The lysosomotropic agents (1) can prevent coronavirus infection by alkalinizing the acid pH in the endolysosomal system; some examples are chloroquine, hydroxychloroquine, azithromycin, artemisinin, two-pore channel antagonists (such as tetrandrine and ned-19) and imatinib. The protease inhibitors/antiviral agents (2) can inhibit the proteolytic cleavage of the spike coronavirus protein, which is necessary for viral entry into host cells; some examples are camostat mesylate, lopinavir, ritonavir, umifenovir and teicoplanin. The third group is composed by PI3K/AKT/mTOR signaling pathways modulators (3), which can modulate intracellular pathways related to autophagy and coronavirus infection; some examples are the rapamycin, wortmannin, the anticoagulant heparin, the glucocorticoid dexamethasone, losartan and cannabidiol. The figures for each chemical structure are from according to Wikimedia Commons (Public Domain).

**Table 1 ijms-22-04067-t001:** Molecular machinery recruited in autophagy initiation.

Acronym	Protein	Function	Ref.
*1. Transcriptional factors*			
TFEB	Transcription factor EB	A master gene regulator of lysosomal biogenesis and autophagy	[54,55]
*2. Initiation of autophagy*			
mTORC1	Mammalian target of rapamycin complex 1	Nutrient sensor and controller of protein synthesis and autophagy	[34]
*3. Upstream regulators of mTORC1*			
AKT	Serine-threonine kinase	Cell growth, proliferation, differentiation and survival signalling	[35,36]
AMPK	Adenosine monophosphate-activated protein kinase	Energy homeostasis signalling	[37]
BCL-2	B-cell lymphoma 2	Regulation of cell death	[40,41]
ERK/MAPK	Extracellular signal-regulated kinase/mitogen-activated protein kinase	Regulation of cell proliferation	[61]
PI3K	Phosphoinositide 3-kinase	Cell growth, proliferation, differentiation and survival signalling	[35,36]
*4. Nucleation and phagophore formation*			
Ambra1	Activating molecule in Beclin-1-regulated autophagy	Positive regulator of Beclin-1-mediated autophagy	[42]
BECN1	Beclin-1	Regulator of autophagic programmed cell death	[40,41]
ULK1	Unc-51 like autophagy activating kinase	Autophagy initiator	[38,39]
*5. Autophagosome formation and elongation*			
Atg	Autophagy-related protein	Factors required for the formation of autophagosomal membranes	[34]
LC3	Microtubule-associated proteins 1A/1B light chain 3A	Autophagosomal marker that mediates the physical interactions between microtubules and components of the cytoskeleton	[43]
p62/SQSTM1	Ubiquitin-binding protein p62/Sequestosome-1	An autophagosome cargo protein that targets and labels other proteins for selective autophagy	[56,70]
Vps34	Vacuolar protein sorting 34	A class III phosphoinositide 3-kinase that acts on vesicle trafficking	[43]
WIPI2	WD repeat domain phosphoinositide-interacting protein proteins	Regulates the assembly of multiprotein complexes	[43]
*6. Autophagosome-lysosome fusion*			
SNAP29	Synaptosome-associated protein 29	Mediates autophagosome-lysosome fusion	[46]
SNARE	N-ethylmaleimide-sensitive factor attachment protein receptor complexes	Vesicle fusion mediator	[46,70]
Stx17	Syntaxin 17	A SNARE like protein that mediates autophagosome-lysosome fusion	[46,70]
VAMP8	Vesicle-associated membrane protein 8	A SNARE like protein that mediates autophagosome-lysosome fusion	[46]

**Table 2 ijms-22-04067-t002:** Potential autophagy-related drugs for SARS-CoV-2 infection.

Drug	Mechanisms	Activity (In Vitro)	Cell Model	Ref.	Current Clinical Trials Number/Phase
**1. Lysosomotropic agents**					
Chloroquine/ hydroxychloroquine	-Prevents endolysosomal acidification; -Blockade of cathepsin activity; -Intracellular retention of ACE2.	SARS-CoV-2	Vero E6	[268]	NCT04341727/Phase 3 NCT04328272/Phase 3
		SARS-CoV-2	Vero E6	[110]	
		SARS-CoV	HEK293E; Vero E6	[117]	
		SARS-CoV	Vero E6	[106]	
Azithromycin	-Acidotropic lipophilic weak base with similar effects to CQ in vitro; -Possible synergy with HCQ for competitive inhibition of SARS-CoV-2 attachment to the host-cell membrane; -Blockade of viral internalization in the early phase of viral infections; -Increases the production of interferon-stimulated genes in rhinoviral infections.	SARS-CoV-2	Vero E6/ In silico	[127,128]	NCT04321278/Phase 3 NCT04381962/Phase 3
		SARS-CoV-2 (presumed)	IB3-1	[126]	
		H1N1	A549	[125]	
		ZIKV	Vero, U87	[124]	
		EBOV	Vero E6	[123]	
		HRV	HBECs	[122]	
Artemisinin and its derivative compounds	-Inhibition of NF-κB; -Chloroquine-like endocytosis inhibition mechanism.	SARS-CoV-2	Vero E6	[144,269]	NCT04387240/Phase 2 NCT04382040/Phase 2
Tetrandrine and ned-19	-Pharmacological inhibition of TPCs; -Inhibition of viral translocation and motility in the endosomal system.	EBOV	HeLa	[158]	NCT04308317/Phase 4
		MERS-CoV	Huh7	[160]	
		HIV-1	U87MG	[270]	
Imatinib	-Inhibitor of ABL-2; -Inhibition of the virion fusion at the endosomal membrane.	SARS-CoV, IBV	Vero E6	[162]	NCT04422678/Phase 3 NCT04346147/Phase 4
		SARS-CoV, MERS-CoV	Vero E6, MRC5, Calu-3, Huh7, BSC1	[164]	
**2. Protease inhibitors/Antiviral agents**					
Camostat mesylate	-Prevents the viral entrance on host cell; -Inhibition of TMPRSS2, a serine protease that cleaves the spike S protein after viral bound to ACE2 receptor.	SARS-CoV-2	Calu-3 and Vero	[101]	NCT04338906/Phase 4 NCT04353284/Phase 2
		SARS-CoV	Caco2	[175]	
Lopinavir/ ritonavir	-Protease inhibitor that prevent viral replication and spread; -Inhibition of protease type 3C; Ritonavir inhibits CYP450.	SARS-CoV-2	Vero E6	[271]	NCT04307693/Phase 2 NCT04372628/Phase 2
		SARS-CoV-2	Vero E6	[272]	
		SARS-CoV-2	Vero E6	[273]	
		SARS-CoV	FRhK-4	[274]	
		MERS-CoV	Vero, Huh7	[275]	
Umifenovir	-Prevents the viral invasion of host cell binding to membrane lipids; -Binds to membrane proteins like clathrin, preventing viral endocytosis through clathrin receptors.	SARS-CoV-2	Vero E6	[271]	NCT04476719/Phase 4 NCT04260594/Phase 4
		LASV, EBOV	HEK293/17 and BSC-1	[182]	
Teicoplanin and others	-Reduces viral invasion by inhibition of cathepsin L activity.	EBOV MERS-CoV SARS-CoV	HEK293, A549 and HeLa	[184]	IRCT20161204031229N3/Phase 3
		SARS-CoV-2	HEK293 and Huh7	[185]	
**3. PI3K/AKT/mTOR modulators**					
Rapamycin	-Inhibition of mTOR pathway;	PEDV	IPEC-J2	[187]	NCT04482712/ Phase 1 and 2
		TGEV	ST, PK15	[188]	
		MERS-CoV	Huh7	[61]	
Wortmannin	-Phosphatidylinositol 3-kinase (PI3K) pathway inhibition;	TGEV	ST, PK15	[188]	N/A
		MERS-CoV	Huh7	[61]	
Heparin	-Inhibition of viral binding with glycosaminoglycans present on the cell surface;	SARS-CoV-2	Vero E6	[197]	NCT04530578/Phase 4
		SARS-CoV	Vero E6	[194]	
		SARS-CoV	HEK293E/ACE2-Myc, Vero E6, Caco-2	[196]	
		HCV	IHH	[192]	
Glucocorticoids	-Glucocorticoid receptor-dependent autophagy activation; -Inhibition of IL-1β, IL-6, IL-8 NF-κB, IFN-β, IFN-λ1 and IFN-γ mediated inflammation;	HCoV-229E	HNE, HTE	[205]	NCT04438980/Phase 3
		HRV	HeLa, Vero E6	[206]	
Losartan	-Inhibition of the AT_1_ receptor;	SARS-CoV	Mice (in vivo)	[276]	NCT04335123/Phase 1
Cannabidiol	-Inhibition of the transmigration of blood leukocytes; -Downregulation of the vascular cell adhesion molecule-1 (VCAM-1), chemokines (CCL2 and CCL5) and the proinflammatory cytokine IL-1β expression; -Attenuation of microglial activation.	HIV	Human (in vivo)	[277,278]	NCT04467918/ Phase 2 and 3
		TMEV	Mice (in vivo)	[247]	

**Table 3 ijms-22-04067-t003:** Therapeutic and toxicological properties for potential autophagy-related drugs against SARSCoV-2 infection according to the PubChem database [279].

Drug	Therapeutic Properties	Toxicological Properties	Compound ID (CID)
Chloroquine/hydroxycloroquine	Malaria and amebiasis treatment and prevention; Rheumatic diseases (i.e., systemic lupus erythematosus and rheumatoid arthritis) treatment.	Corneal deposits, posterior subcapsular lens opacity, ciliary body dysfunction, retinopathy and cardiac rate changes.	2719 3652
Azithromicin	Mild-to-moderate Gram positive (i.e., staphylococci) and Gram negative (i.e., *Mycoplasma pneumonia*) bacterial infections treatment; Protozoan infections (i.e., *Toxoplasma gondii* and *T. cryptosporidia* ) treatment.	Hepatotoxicity, nephrotoxicity and severe cutaneous reactions (i.e., erythema multiforme and toxic epidermal necrosis).	447043
Artemisinin	Leishmaniasis and Malaria treatment.	Sedative in rodent models but no significant toxicity has been reported in humans; Cardiotoxicity and QT interval prolongation	68827
Tetrandrine	Adjunctive therapy to chemotherapy in various cancer types with multiple drug resistance; Antiviral activity against Ebola virus; Anti-inflammatory and antifibrogenic actions in lung silicosis, liver cirrhosis, and rheumatoid arthritis.	Local pain, phlebitis and tissue irritation; Mild and transient hearing loss, peripheral neuropathy, cerebellar toxicity and cardiotoxicity.	73078
Imatinib	Treatment of chronic myeloid leukemia, lymphoblastic leukemia, myelodysplastic/myeloproliferative diseases, aggressive systemic mastocytosis, hypereosinophilic syndrome and/or chronic eosinophilic leukemia, dermatofibrosarcoma protuberans and malignant gastrointestinal stromal tumors.	Edema, nausea, vomiting, muscle cramps, musculoskeletal pain, diarrhea, rash, fatigue and abdominal pain.	5291
Camostat mesylate	Chronic pancreatitis.	N/A	5284360
Lopinavir/ritonavir	Antiretroviral activity against Human Immunodeficiency Virus-1 (HIV-1).	Atrioventricular block, cardiomyopathy, lactic acidosis, and acute renal failure.	11979606
Umifenovir	Broad-spectrum antiviral against influenza and other respiratory viral infections, *Flavivirus*, Zika virus, foot-and-mouth disease, Lassa virus, Ebola virus and herpes simplex.	Chronic administration of doses 10–50 times higher than the therapeutic human dose resulted in no pathological changes to animal subjects.	131411
Teicoplanin	Antibiotic against pseudomembranous colitis and *Clostridium difficile*.	Change in auditory acuity and ototoxicity.	133065662
Rapamycin	Potent immunosuppressant with both antifungal and antineoplastic properties.	Peripheral edema, hypercholesterolemia, abdominal pain, headache, nausea, diarrhea, chest pain, stomatitis, nasopharyngitis, acne, upper respiratory tract infection, dizziness and myalgia.	5284616
Heparin	Anticoagulant; Antitumoral agent with angiogenesis inhibiting properties.	Heparin-induced thrombocytopenia, which may progress to arterial thrombosis, gangrene, stroke, myocardial infarction; Spontaneous fractures and osteoporosis.	772
Dexamethasone (glucocorticoid)	Anti-inflammatory and immunosuppressive agent for a number of endocrines, rheumatic, collagen, dermatologic, allergic, ophthalmic, gastrointestinal, respiratory, hematologic, neoplastic, edematous and other conditions.	Chronic high doses of glucocorticoids can lead to the development of cataract, glaucoma, hypertension, water retention, hyperlipidemia, peptic ulcer, pancreatitis, myopathy, osteoporosis, mood changes, psychosis, dermal atrophy, allergy, acne, hypertrichosis, immune suppression, decreased resistance to infection, moon face, hyperglycemia, hypocalcemia, hypophosphatemia, metabolic acidosis, growth suppression and secondary adrenal insufficiency.	5743
Losartan	Antihypertensive able to reduce the risk of stroke in patients.	Hypotension, tachycardia, or bradycardia due to vagal stimulation.	3961
Cannabidiol	Analgesic, anticonvulsant, muscle relaxant, anxiolytic and antipsychotic agent; Treatment of rare forms of refractory epilepsy syndromes.	Sedation, somnolence and fatigue; Drug-drug interactions and hepatic abnormalities.	644019

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
