# Peer review of "Pharmacological Modulators of Autophagy as a Potential Strategy for the Treatment of COVID-19"

_ijms, 2021, doi:10.3390/ijms22084067_

Round 1

Reviewer 1 Report

The authors tried to review relationships among COVID-19 treatment, various types of the drugs and autophagy. I felt that their viewpoints was good and interesting. However, some concerns were found.

Major

1. They missed a prompt definition of autophagy. Is autophagy related all biological events? The authors should review based on the authentic definition of autophagy.  

2. I felt that there were too many authors.     

Minor

Table should be more refined (positions etc.).

Reviewer 2 Report

The manuscript "Pharmacological Modulators of Autophagy as a Potential Strategy for the Treatment of COVID-19" by Gustavo Pereira and colleagues describes with great detail the current knowledge on how SARS-CoV-2 exploits innate cellular pathology and how this can be used as the molecular basis for a specific pharmacological strategy. The review is well written, and my observations are mostly regarding the current lack of details on some of the topics discussed.

1) Given the unprecedenteed fast pace of the literature on the subjects of COVID-19 and SARS-CoV-2, I would suggest the authors do a quick literature review and update before final submission, and beyond my suggestions below.

2) The authors should be precise in terms of which viral components are acting in the autophagy exploitation model. In other words: is the Spike protein the only component that is involved in triggering the endocytosis? Are there evidences of other surface proteins that can interact with the endosomal pathway components (both extra- and intra-cellular)? The authors should look at existing data that describe the sars-cov-2/human interactome, to highlight other key SARS-CoV-2 components beyond Spike, which will have potential ramifications also in target definition and drug design. Some example literature on the SARS-CoV-2/human interactome: PMID 33349665, PMID 32511329, PMID 32244779.

3) The use of HCQ in anti-SARS-CoV-2 therapy is very controversial at the moment. The authors do an excellent work in objectively highlighting all the existing literature on the subject (page 6). An interesting new brief commentary on the International Journal of Antimicrobial Agents has been published only a few days ago, PMID 33408026, showing how the apparent efficacy of HCQ is often associated with poor study designs.

4) There is more to say on the role of autophagy in other RNA viruses, e.g. PMID 19066474, PMID 32943650, PMID 31861933. Also, it should be interesting to expand on whether there are differences in autophagy between major viral families (e.g. DNA, RNA, retroviruses). In other words, whether autophagy, and this review, could be generalized as a strategical framework for other viruses as well.

5) There is no mention, throughout the paper, of the dangers given by future mutations of SARS-CoV-2. New clades/variants/strains may arise, changing the molecular properties of the virus, how it can trigger autophagy, and ultimately if pharmacological strategies (drugs) repurposed or designed against autophagy will be potentially affected by viral evolution. The necessity to keep monitoring mutations, maybe in key viral components involved in autophagy (see also my point 2) should be mentioned. A recent review on ways to keep track of SARS-CoV-2 mutations and their pharmacological effects is e.g. PMID 33057582

6) Autophagy in viral entry and infection has multiple roles, sometimes even anti-viral, and the authors should mention the existing dichotomy, e.g. PMID 24709646

Round 2

Reviewer 1 Report

The authors addressed to scientific concerns. However, I could not understand each author's contribution in the study. Thus, more detailed roles of each author should be described in Author Contribution section. 

Reviewer 2 Report

The authors provide an update manuscript that excellently addresses all the comments, by expanding, updating and deepening some aspects of autophagy in several contexts.

I have only a minor point. "Combat the COVID-19" (Abstract, line 16) does not require the article "the". The authors write it correctly elsewhere in the text, e.g. Page 1 line 47, Page 1 line 52 and page 2, line 62. Elsewhere, the article is appropriate, e.g. "the COVID-19 pandemic" (Page 7, line 300), because it refers to the noun "pandemic".
